# Dopamine Photochemical Behaviour under UV Irradiation

**DOI:** 10.3390/ijms23105483

**Published:** 2022-05-13

**Authors:** Alexandra Falamaş, Anca Petran, Alexandru-Milentie Hada, Attila Bende

**Affiliations:** 1National Institute for Research and Development of Isotopic and Molecular Technologies, Donat Street, No. 67-103, 400293 Cluj-Napoca, Romania; alexandra.falamas@itim-cj.ro (A.F.); anca.petran@itim-cj.ro (A.P.); 2Nanobiophotonics and Laser Microspectroscopy Center, Interdisciplinary Research Institute in Bio-Nano-Sciences, Babes-Bolyai University, 42 T. Laurian Str., 400271 Cluj-Napoca, Romania; alexandru.hada@ubbcluj.ro; 3Faculty of Physics, Babes-Bolyai University, 1 M. Kogalniceanu Str., 400084 Cluj-Napoca, Romania

**Keywords:** dopamine, time-resolved fluorescence, excited state lifetime, TDDFT, DLPNO-STEOM

## Abstract

To understand the photochemical behaviour of the polydopamine polymer in detail, one would also need to know the behaviour of its building blocks. The electronic absorption, as well as the fluorescence emission and excitation spectra of the dopamine were experimentally and theoretically investigated considering time-resolved fluorescence spectroscopy and first-principles quantum theory methods. The shape of the experimental absorption spectra obtained for different dopamine species with standard, zwitterionic, protonated, and deprotonated geometries was interpreted by considering the advanced equation-of-motion coupled-cluster theory of DLPNO-STEOM. Dynamical properties such as fluorescence lifetimes or quantum yield were also experimentally investigated and compared with theoretically predicted transition rates based on Fermi’s Golden Rule-like equation. The results show that the photochemical behaviour of dopamine is strongly dependent on the concentration of dopamine, whereas in the case of a high concentration, the zwitterionic form significantly affects the shape of the spectrum. On the other hand, the solvent pH is also a determining factor for the absorption, but especially for the fluorescence spectrum, where at lower pH (5.5), the protonated and, at higher pH (8.0), the deprotonated forms influence the shape of the spectra. Quantum yield measurements showed that, besides the radiative deactivation mechanism characterized by a relatively small *QY* value, non-radiative deactivation channels are very important in the relaxation process of the electronic excited states of different dopamine species.

## 1. Introduction

Research on dopamine (DA) is of ongoing interest due to its wide range of applications in many fields, such as biotechnology, nanotechnology, and advanced materials science. The successive oxidation processes of dopamine lead relatively easily to a more complex chemical structure known in the literature as polydopamine (PDA) [1,2,3]. PDA is an important polymer, especially for coating various surfaces on almost all materials, with conformal layers of adjustable thickness from a few to about 100 nm [3,4,5,6,7]. Detailed structural studies have revealed a heterogeneous form of PDA with respect to the basic dopamine unit, implying that PDA cannot be described by a single, well-defined structure [1,8,9]. Non-invasive spectroscopy techniques such as FT-IR [10], UV [11], EPR [10], or XPS [1,11] have been able to confirm several structural features of PDA, but do not unambiguously reveal the polymer structure. Several chemical units based on catechol-, quinone-, indole-, or indoline-type units have been identified in the primary structure of PDA, but the supramolecular structure, more specifically how these units are coupled to form a polymer, is only partially elucidated [9,12,13]. Although polymerisation is difficult to control due to the complexity of the oxidation process, it is possible to influence the process in a controlled way under certain conditions [14,15,16,17]. The existence of these properties, on the other hand, opens up the possibility of developing so-called stimuli-sensitive PDA-based smart materials [18].

Although PDA shares many basic properties with amorphous organic semiconducting polymers such as melanin [19], many of its physicochemical properties, in particular its photochemical behaviour, are still not fully understood. Several optical properties of PDA have been investigated in various experimental studies [20,21,22,23,24,25,26,27], but without detailed knowledge of the photochemical behaviour of individual PDA units, in particular dopamine, it is difficult to interpret the absorption or photoluminescence spectra properly. Dopamine’s absorption and fluorescence spectra were experimentally studied and discussed by Mabuchi et al. [28] and Wang et al. [29]; however, a thorough analysis of the characteristics of these spectra has not yet been published. The difficulty is that the molecular structure of dopamine is very diverse, ranging from the standard chemical form, the zwitterion version [30], to the quinone configuration, all of which have different geometries in either the liquid or solid state [30,31,32], and these configurations in turn have also different spectroscopic fingerprints [33], which are very dependent on the environmental conditions [16,17].

The main goal of the present investigation is to give a detailed description of the dopamine molecular structure including their light absorption and emission properties, as well as electronic excited state lifetimes using time-resolved spectroscopy and theoretical molecular modelling techniques.

## 2. Materials and Methods

### 2.1. Samples’ Preparation

All reagents were commercially available and used without further purification. The buffer solutions were prepared based on the standard procedures reported for air oxidation in acidic (NaAc 50 mM, pH = 5.5) or basic (TrisCl 10 mM, pH = 8) conditions. UV–Vis and fluorescence spectra were measured at specific dopamine concentrations as follows: concentrated −0.7 mg/mL DA and diluted −0.05 mg/mL DA. Absorption spectra were recorded after the first 5 min of the sample preparation.

### 2.2. Steady-State UV–Vis Absorption and Fluorescence Spectroscopy

The UV–Vis absorption spectra were recorded using a double-beam Jasco V550 UV–VIS spectrophotometer in the 200–800 nm spectral range with a 2 nm resolution, using 1 mm-length quartz cells. Baseline corrections were run successively in air and the solvent. The fluorescence emission and excitation spectra of dopamine aqueous solutions were acquired using a Jasco FP-6500 spectrofluorometer, equipped with a Xe lamp of 150 W, 1800 lines/mm monochromator, and a 1 nm spectral resolution. The samples were measured in 3 mm optical path quartz cuvettes. The quantum yield and fluorescence spectra of dopamine solutions at pH 5.5, respectively pH 8, were acquired using an FP-850 NIR spectrofluorometer with a 1 nm resolution for both excitation and emission, equipped with a 1800 lines/mm monochromator and an 100 mm integrating sphere (ILF-835).

### 2.3. Time-Resolved Fluorescence Spectroscopy

The fluorescence lifetimes were investigated by pumping the dopamine solutions with 280 nm laser pulses. The pump was generated by frequency doubling the 560 nm second harmonic of the 1120 nm idler beam obtained using an optical parametric amplifier (OPA) (Orpheus, Light Conversion). The OPA was pumped by a 1030 nm Yb:KGW pulsed laser (Pharos, Light Conversion) with a 170 fs pulse duration and an 80 kHz repetition frequency. The fluorescence kinetics was investigated using a time-correlated single photon counting (TCSPC) setup (Chimera, Light Conversion). The samples were excited with vertically polarized light, and the fluorescence emission was collected through a long-pass filter suitable for the sample emission and an emission polarizer set at the magic angle (54.7° relative to the polarization of the pump pulse). The fluorescence signal was focused on the entrance slit of a double monochromator and detected using a single-photon-sensitive photomultiplier (Becker&Hickl PMC-100-1 standard). The FWHM of the instrument response function (IRF) was 340 ps, and the IRF was measured on each day of the experiment. For the TCSPC measurements, the average power of the pump beam was 3 to 10 mW at the probe location. For fitting the kinetic curves, we employed the reconvolution method available in the Easy Tau 2.0 software from PicoQuant with both one and two exponentials in order to determine the best-fitting parameters.

The absolute quantum yield (*QY*) was calculated using the FP-8600 spectrofluorometer from JASCO equipped with an ILF-835 integrating sphere accessory. The measurements were performed in a quartz cuvette of 3 nm thickness under 280 nm excitation, while the quantum yield was obtained via the spectra manager software provided by JASCO. The *QY* calculations are based on the following equation:(1)$$QY=\frac{E_{2}}{L_{1}-L_{2}},$$
where *E*${}_{2}$ is the area of the fluorescence emission band, *L*${}_{1}$ is the area of the excitation band, and *L*${}_{2}$ is the area of the excitation band that was not absorbed by the probe.

### 2.4. Theoretical Methods

The equilibrium geometries and normal mode vibrational frequencies were obtained in the framework of density functional theory (DFT) considering the B97X exchange-correlation (XC) functional [34] combined with the D3-type empirical dispersion correction scheme [35,36] and applying the minimally augmented [37] ma-def2-TZVPP triple- basis set of the Karlsruhe group [38] as implemented in the Orca program suite [39,40]. The electronically excited state calculations were computed using the time-dependent version of the same DFT framework considering the Tamm–Dancoff approximation (TDA) approximation [41]. The RIJCOSX approximation [42] designed to accelerate Hartree–Fock and hybrid DFT calculations were considered together with the Def2/J [43] auxiliary basis set for Coulomb fitting and the def2-TZVPP/C [44] auxiliary basis set for correlation fitting in the case of TD-DFT calculations. The electronically excited states were also computed considering the DLPNO-STEOM-CCSD coupled-cluster method [45,46,47,48] implemented in the same Orca package. The theoretical prediction of the fluorescence rate is based on the path integral approach to the dynamics by solving Fermi’s Golden Rule-like equation including vibronic coupling in forbidden transitions (the so-called Herzberg–Teller effect (HT)) and Duschinsky rotations between modes of different states [49]. The solvent environment of water was taken into account through the conductor-like polarizable continuum (CPCM) model [50].

## 3. Results and Discussion

### 3.1. UV–Vis Absorption and Fluorescence Spectra

UV–Vis absorption, as well as the fluorescence excitation and emission spectra were recorded for aqueous DA solutions with different DA concentrations (see Figure 1). The most concentrated DA solution was taken as 1 mg DA in 0.5 mL, while the most diluted as 1mg DA in 30 mL solution, all of them prepared at a neutral pH value (between 7.0–7.2). UV absorption spectra of a solution with a low and a high DA concentration, as shown in Figure 1a, present similar spectral characteristics with peak maxima at 280 nm and 218 nm. Furthermore, analysing Figure 1b, it can be observed that all aqueous DA solutions showed the same fluorescence emission maxima centred at 317 nm. The fluorescence excitation spectra, however, presented a more complex behaviour, which changed with increasing the concentration of the DA solution. At high concentrations (1 mg/0.5 mL and 1 mg/4 mL), two main bands were observed located at 258 and 290 nm, which merged into a peak at 280 nm with decreasing the concentration. This effect can be mainly assigned to the so-called inner filter effect [51], as indicated by the corresponding emission spectra, which presented decreased intensity at the highest concentrations (1 mg/0.5 mL, respectively 1 mg/4 mL) compared to the 1 mg/8 mL concentration, but the presence of multiple DA structural configurations or DA aggregation for highly concentrated DA cannot be ruled out.

As regards the pH dependence of UV–Vis absorption, as well as the fluorescence excitation and emission spectra, several solutions of diluted DA with pH ranking from 5 up to 12 were also prepared, and their spectra were recorded (see in Figure 2). As can be inferred from the shapes of the absorption spectra (see Figure 2a), there was no significant change in the spectral curves for pH 5 to 7.8, only a slight change in peak magnitude. For pH 8.5 and 9, a small shoulder of the spectral curve appeared near at 300 nm and another small shoulder at 250 nm. However, at pH 10.5 and 12, spectral changes were seen that indicated a change in the molecular structure of DA, as a new peak appeared at 430–450 nm, which was not present at the previous pH values. This structural change could mean either the complete deprotonation (appearance of o-quinone form) of oxygens or the closure of the NH${}_{2}$ fragment into a five-membered ring attached to the existing six-membered ring (appearance of 2,3-dihydro-indolo-5,6-quinone) [52]. These new molecular structures, however, will not be discussed further because they would not reveal the photochemical behaviour of dopamine. Thus, solutions above pH 9 were not analysed because they would contain less photochemical fingerprints of dopamine. The spectral differences observed for concentrated and diluted solutions suggest that not only one structural configuration is present, but also several geometric conformations. It is well known that in the solid phase, DA takes the zwitterionic [30] form, where the two OH fragments are replaced by the OHO${}^{-}$ form, while instead of the NH${}_{2}$ fragment, the NH${}_{3}^{+}$ form appears. Furthermore, when the pH of the solution is modified to either low or high pH values, not only the zwitterionic form is possible, but also the protonated and deprotonated conformation [52,53,54,55]. In order to obtain a more accurate picture of the distribution of the different dopamine conformations according to the pH, a qualitative estimation based on the pKa analysis was performed [56,57]. Detailed results can be found in the Appendix A (see Appendix A). Correspondingly, at low pH, the most probable DA configuration is the NH${}_{3}^{+}$ protonated fragment, but at pH ≥ 8, both the normal DA configuration and the zwitterion forms appear. At pH ≥ 9, the singly deprotonated forms are also present. A similar analysis based on pKa values predicts a similar configurational distribution for dopamine in aqueous solution [54].

Accordingly, a total of four different conformations of dopamine were considered. Namely, the standard form, which we denote as DA, the zwitterion form as DA${}^{zw}$, the protonated (NH${}_{3}^{+}$) form as DAH${}^{+}$, and the deprotonated (OHO${}^{-}$) form as DAH${}^{-}$, respectively. For their chemical structures, see Figure 3a–d. To properly interpret the differences in the UV absorption spectrum, theoretical ab initio calculations were performed. Accordingly, the four equilibrium geometry conformations were optimized considering the ωB97X-D3/ma-def2-TZVPP level of theory, while their low-lying (up to ten) electronic excited states were computed considering the DLPNO-STEOM-CCSD/ma-def2-TZVPP level of theory. The first ten electronic transitions and their oscillator strengths found for the DA, DA${}^{zw}$, DAH${}^{+}$, and DAH${}^{-}$ equilibrium geometries are collected in Table 1. If we compare the experimental UV absorption and fluorescence excitation spectra with the theoretically obtained energies of the electronic excited states, we found that the corresponding excitation energies of DAH${}^{+}$ (S${}_{1}$ = 270 nm and S${}_{2}$ = 222 nm) are in good agreement with the spectra of the absorption and fluorescence excitations with a peak at 280 nm. However, it is possible to obtain a better match by explicitly considering 1–3 water molecules in the immediate vicinity of the DA [55]. It was also observed that at higher pH values, a slight shoulder appears on the spectral curve, and the peaks of the curves show a slight red shift, which could be explained by the presence of zwitterion species based on the determination of the theoretical energies.

As far as the fluorescence emission spectra are concerned, it can be observed that the frequency of the maximum of the spectrum does not depend on whether the DA solution was prepared in concentrated or diluted form. For each DA geometry conformation, a very broad peak was found for the fluorescence spectrum with a maximum around 317 nm (see the dashed lines in Figure 1b and the peaks from the right side of Figure 2b) with a full-width at half-maximum (FWHM) given by the vibrational sublevels of around 40 nm. At the same time, the experimental data and theoretical calculations for the fluorescence spectra are not expected to match, as well as those obtained for the absorption spectra, since it was demonstrated that the accuracy of the geometry optimization of the electronic excited states by the TDDFT method is poorer than that of the ground state obtained by the classical DFT method [58]. The geometries of the first electronic excited states for the DAH${}^{+}$, DA, and DA${}^{zw}$ conformations were optimized considering the ωB97X-D3/ma-def2-TZVPP level of theory, while their low-lying electronic excited states were computed considering the DLPNO-STEOM-CCSD/ma-def2-TZVPP level of theory. Accordingly, the first excited state energy (S${}_{1}^{opt}$) found for DAH${}^{+}$ was 300 nm, for the standard DA conformation was 302 nm, while that for DA${}^{zw}$ was 334 nm. It can be seen that these theoretical values for fluorescence emission lie within the interval covered by the corresponding experimental spectrum.

As for the influence of the pH on the absorption spectra, we investigated the optical properties of DA solutions at various pH values in the 5–12 interval (see Figure 2 and Figure 4). At pH 5, the spectrum presented the 280 nm band and an intense peak at 209 nm. With increasing the pH of the solution, several spectral changes were identified. First, a shoulder appeared around 300 nm for pH values around 8–9. At pH above 10, a visible absorption band developed around 440 nm. Additionally, at high pH values, a third band developed in the UV region, as well as around 238 nm. The 280 nm peak presented a 13 nm red-shift when the pH was modified from 5 to 12. Moreover, the absorbance of the solutions increased with the pH, as well. The emission spectra corresponding to each pH value looked similar to one another, except for a 4 nm red-shift observed between the lowest pH and the pH 10.5 characteristic spectra. Additionally, a four-times decrease of the emission intensity was calculated. At pH 12, the 317 nm peak disappeared and a weak emission band appeared around 470 nm. The acquisition of the DA emission spectra at high pH solutions is hindered by its rapid oligomerization [15]. Similarly, the excitation spectra presented a 5 nm blue-shift of the 280 nm band, accompanied by a four-times intensity decrease when going from pH 5 to pH 10.5.

In order to better understand the photochemical behaviour at lower and higher pH values, a specific comparison was made for pH 5.5 and pH 8. Both absorption and excitation spectra showed a peak around 280 nm that was slightly shifted to higher wavelengths. It was also observed that the intensity of the absorption peak increased slightly, while the fluorescence excitation intensity decreased by a factor of about 8 for the pH 8 solution compared to the pH 5.5 solution. Similarly, the emission spectra also showed the same shifting of the fluorescence maximum (316 nm at pH 5.5, respectively 318 nm at pH 8) intensity decrease for the pH 8 solution in comparison to the pH 5.5 solution. At first glance, the two absorption spectra (black line in Figure 4) hardly differ, yet it is thought that the geometries that generate these spectra are different depending on the pH values. That is, at pH 5.5, the DAH${}^{+}$ geometry is dominant, whereas at pH 8, the DAH${}^{zw}$ geometry also could have a significant contribution. The electron transitions determined by the DLPNO-STEOM-CCSD/ma-def2-TZVPP method seem to confirm this finding. For pH 5.5, the S${}_{0}$→S${}_{1}$ transition of DAH${}^{+}$ is dominant with an energy value of 270 nm, while for pH 8, the S${}_{0}$→S${}_{2}$ transition of the DAH${}^{zw}$ geometry is the most intense, corresponding to a value of 286 nm. The S${}_{0}$→S${}_{1}$ transition is also likely because around 290 nm, the spectrum has a slight shoulder, which supports the presence of the S${}_{0}$→S${}_{1}$ transition, but with a small peak intensity. This was confirmed also by theoretical calculations. Namely, in the case of the DAH${}^{zw}$ geometry, the oscillator strength of the S${}_{0}$→S${}_{1}$ transition was only 0.0014, while that of the S${}_{0}$→S${}_{2}$ transition was 0.0803, which is almost two orders of magnitude higher than the previous one. The fluorescence emission spectra of diluted aqueous DA solutions showed similar shapes for the two pH values, except that at pH 8, a slight red-shift was observed, together with a decrease in intensity, compared to the emission spectra recorded from the pH 5.5 solution. Similarly, the concentration dependence of the fluorescence intensity results was found also by Wang et al. [29]. Theoretical calculations performed in the case of the DAH${}^{+}$ conformation for the equilibrium geometry of the first electronic excited state gave an energy value of 300 nm, while for the DAH${}^{zw}$ conformation 334 nm. The two frequency values showed a shift with respect to each other, but they were still close to the experimentally measured fluorescence peak values (316 and 318 nm, respectively).

In short, one can say that the fluorescence excitation spectrum of dopamine can be considered not so much pH-dependent as concentration-dependent, since the shape of the spectra differed much more when comparing concentrated (mixed standard and zwitterionic form) and diluted forms than was observed when varying the pH of the solvent. However, it should also be noted that, based on theoretical calculations, S${}_{0}$ to S${}_{2}$ excitation is more likely than the S${}_{0}$→S${}_{1}$ transition in the case of a concentrated or higher pH mixture due to more efficient absorption and electronic transition. This also means a bit more complex deactivation path for the concentrated (DA${}^{zw}$ case) or pH = 8 diluted (the most likely case is DA${}^{zw}$, but DAH${}^{-}$ can also occur) DA aqueous solutions.

### 3.2. Fluorescence Lifetimes and Quantum Yield

Time-correlated single-photon counting (TCSPC) was applied to determine experimentally the fluorescence lifetimes of dopamine in buffer solutions at pH 5.5 and 8, respectively. Figure 5 presents the three-dimensional data carpets acquired from concentrated and diluted dopamine solutions at pH 5.5 obtained following 280 nm pulsed laser excitation. The data carpets were recorded in the 310–350 nm spectral range with a 2 nm step size. The data show an intense, short-lived fluorescence signal. The kinetic traces do not exhibit dissimilarities from one emission wavelength to another. Moreover, the integrated emission spectra presented overlapped on the data carpet are similar to the ones obtained from steady-state fluorescence spectroscopy. The kinetic traces recorded at 322 nm emission wavelengths from both concentrated and diluted solutions are given in Figure 5c. Slight variations between the two kinetic traces can be observed, especially in the last part of the curve.

When the obtained fitting curves were analysed together with the corresponding kinetic curves of the signal from the concentrated dopamine solution, it was observed that the curve obtained with the one-exponential fitting did not cover the tail of the 322 nm kinetic curve, in contrast to the curve obtained with the two-exponential fitting. The residuals for the fitting procedure with the two-exponential function showed an improvement in the time scale corresponding to the tail of the kinetic curve. When only one exponential function was employed, the resulted fluorescence lifetime was $\tau_{1}$ = 0.89 ± 0.01 ns; however, the $\chi^{2}$ parameter was 2.7. For the case when two exponential functions were employed, we obtained time components of $\tau_{1}$ = 0.91 ± 0.01 ns and of $\tau_{2}$ = 4.86 ± 0.4 ns. The $\chi^{2}$ parameter was significantly improved to 1.9. Even if a better-quality fitting was obtained, it should be mentioned that the averaged lifetime component was 0.91 ± 0.02 ns, due to the fact that the longer fluorescence lifetime had a relative amplitude of only 0.5%. Still, basing our assumption on the previous investigations, we expect that the concentrated solution at pH 5.5 contains both the zwitterionic DA${}^{zw}$ conformation, as well as the deprotonated form, namely DAH${}^{+}$. In the case of the diluted pH 5.5 dopamine solution, one exponential function was enough to obtain a proper fit of the kinetic curves. The resulting lifetime component had a value of 0.93 ± 0.07 ns. The averaged lifetime components calculated for the entire emission spectral ranges, together with the average amplitude lifetimes, are given in Table 2.

Moving on to the dopamine solution at pH 8 (see Figure 6), a decreased intensity data carpet was acquired in the same conditions as those employed in the case of the pH 5.5 solution. The decrease of the signal is especially evident for the diluted pH 8 solution. The kinetic traces required two exponential functions for a proper fit to be obtained. The dominant $\tau_{1}$ lifetime was 0.89 ± 0.01 ns for the concentrated solution and slightly shorter (0.71 ± 0.10 ns) for the diluted solution. The $\tau_{2}$ fluorescence lifetime specific for the tail of the kinetic curve was around 3 ns (2.98 ± 0.20 ns for concentrated and 3.23 ± 0.24 ns for diluted solution). The averaged amplitude lifetime components presented also similar values for both solutions, with a slight decrease in the case of the concentrated DA solution.

For the dopamine solution at pH 9, the dominant fluorescence lifetime remained $\tau_{1}$ (0.69 ± 0.02 ns for the concentrated and 0.64 ± 0.10 ns for the diluted case), but the values decreased a bit as compared to the results obtained at pH 8. It is also important to note that the contribution of the $\tau_{2}$ component to the average lifetime increased with increasing pH, which means that other configurations start to appear more intensively at a higher pH compared to the structure of DAH${}^{+}$, which is the only configuration present at low pH.

The theoretical estimation of different fluorescence lifetimes for the four DA conformations was performed at the ωB97X-D3/ma-def2-TZVPP level of theory by computing the vibrational normal modes both for ground and first excited states. Three different levels of approximation were considered [59,60]: *(i)* the so-called Franck–Condon (FC) approximation, where the electronic transition is considered “vertically”, meaning that the electronic transition is most likely to occur without changes in the positions of the nuclei in the molecular entity and its environment; *(ii)* the Herzberg–Teller (HT) approximation, where, besides the “vertical” transition, also the vibronic coupling is taken into account; *(iii)* the Duschinsky (D) rotations between normal modes of different states is also considered.

The fluorescence lifetimes calculated as the inverse of the radiative transition rate for the four DA species considering the three *(i)–(iii)* levels of approximation are collected in Table 3. Furthermore, the transition rate calculation makes it possible to discriminate between the FC and HT percentage contributions. Accordingly, the individual percentage contribution of the FC and HT contributions to the fluorescence transition rate is also shown in the table (see the form of (x%,y%)). As can be seen in Table 3, for DA and DAH${}^{+}$, the vibronic contributions did not change the lifetimes in magnitude; only the numerical values varied slightly. This was confirmed by the percentage contributions, with the FC approximation dominating in both cases (81% and 78%, respectively). In the other two cases (DA${}^{zw}$ and DAH${}^{-}$), the situation was the other way round, only the vibrational effects made the radiative transition possible; the “vertical” transition was 2–3 orders of magnitude slower. This is clearly due to the fact that the S${}_{1}$→S${}_{0}$ transition dipole moment is very small in these two cases, which makes the transition almost forbidden. It is also clear that for the DA${}^{zw}$ and DAH${}^{-}$ forms, the C-O${}^{-}$ bond can form a tighter hydrogen bond with the water molecule in close proximity to this bond. Accordingly, in both cases, a water molecule was also placed next to the C-O${}^{-}$ bond, and the transition rate was recalculated. The values obtained for the transition rate under these conditions were higher and the lifetimes shorter: τ(DA${}^{zw}$) = 0.84 ns and τ(DAH${}^{-}$) = 0.66 ns.

As far as the correlation between experimental and theoretical results on fluorescence lifetimes is concerned, a very good agreement in terms of the numerical value and magnitude was found. Furthermore, it is possible to make a correspondence between lifetimes and different DA geometries. Accordingly, the slightly longer $\tau_{2}$ lifetimes obtained experimentally are more likely to be a contribution of standard (or DA) and protonated (or DAH${}^{+}$) geometries, while the shorter $\tau_{1}$ lifetimes may be a characteristic of zwitterion (or DA${}^{zw}$) and deprotonated (or DAH${}^{-}$) geometries.

As was shown especially for the DA${}^{zw}$ and DAH${}^{-}$ cases, electron-vibrational coupling is very important for radiative relaxation. The question arises which normal modes are the ones that exhibit stronger coupling and, thus, contribute to the fluorescence phenomenon. The fluorescence rate constant is strongly related to the Huang–Rhys factor (HRF) and reorganization vibrational energy [61,62]. Accordingly, vibrational normal modes having larger (≥0.1) HRFs were identified. The vibrational frequencies, the HRFs, and the reorganization energies, as well as the graphics of the vibrational amplitudes of the corresponding normal modes can be seen in the SM file (see Table S1). The strongest vibronic coupling was obtained for normal mode frequencies of 749 cm${}^{-1}$ and 1355 cm${}^{-1}$ for DA, 1437 cm${}^{-1}$ for DA${}^{zw}$, 753 cm${}^{-1}$ and 1366 cm${}^{-1}$ for DA${}^{+}$, and 1547 cm${}^{-1}$ for DA${}^{-}$, respectively. In all four cases, we saw a delocalized normal mode vibration, where the aromatic ring was deformed together with the OH groups (or C=O bond) stretching.

It is well known that the process of the deactivation of the electronically excited state can occur not only radiatively, through the phenomenon of fluorescence, but also non-radiatively. An integrating sphere was used to calculate, for the first time, the fluorescence *QY* of dopamine in buffer solutions at pH 5.5 and 8, respectively, under 280 nm excitation (see Table 4). The *QY* presented a decrease of 1.7-times for pH 8 compared to pH 5.5 solution, from 3.4% to 2%. Taking into consideration that at pH 8, the absorption decreased in intensity about four times, while the fluorescence suffered a higher than four-fold intensity decrease in comparison to the solution at pH 5.5, the *QY* values are in direct correlation with the aforementioned results. Moreover, the diluted solution at pH 5.5 exhibited no *QY* modifications, while a slight variation was observed at pH 8 by an increasing of 0.4% in the diluted case. Therefore, the *QY* of dopamine fluorescence shows both pH-dependent and, surprisingly at higher pH, concentration-dependent behaviour. In general, it can be said that, besides the radiative deactivation mechanism characterized by a relatively small *QY* value, non-radiative deactivation channels are very important. It is also true that these non-radiative deactivation pathways are present for all four DA geometries, with a slightly higher extent for pH 8. In order to unravel the nature of non-radiative deactivation mechanisms, a much more detailed and thorough theoretical investigation would be needed, where the different pathways characteristic for the internal conversions are discussed in detail.

## 4. Conclusions

In this work, the photochemical behaviour of different dopamine species (standard, zwitterionic, protonated, and deprotonated geometries) were investigated considering steady-state and time-resolved spectroscopy and first-principles quantum theory methods. Different peaks found in the experimental absorption and fluorescence excitation spectra were attributed to electronic transitions occurring in different dopamine species based on DLPNO-STEOM coupled-cluster calculations. The results show that not all spectra can be attributed to only one or another dopamine species, but there are cases of concentrated and higher pH dopamine solutions where a mixture of different species of protonated, standard, and zwitterionic geometries give the final spectra. On the other hand, it was found that the fluorescence excitation spectrum of dopamine can be considered not so much pH-dependent as concentration-dependent. The fluorescence lifetimes measured for concentrated or diluted, as well as for solutions with a low or high pH using the time-correlated single-photon counting method were in the range of 1 ns, values that were confirmed by the results of theoretical calculations based on the path integral approach. The used approximation levels for the theoretical estimation of the fluorescence lifetime also proved that the fluorescence deactivation is not entirely based on Franck–Condon effects; vibronic couplings covered by Herzberg–Teller effects also have a significant contribution to the fast radiative relaxation. Furthermore, quantum yield measurements showed that, besides the radiative deactivation mechanism characterized by a relatively small *QY* value, non-radiative deactivation channels are very important in the relaxation process of the electronic excited states of different dopamine species.

## Figures and Tables

**Figure 1 ijms-23-05483-f001:**
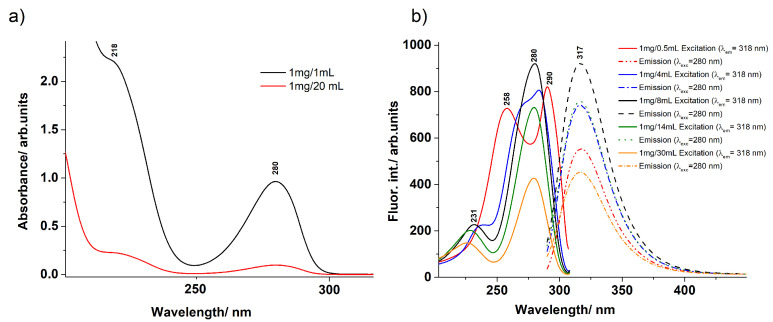
(**a**) The experimental UV absorption spectra obtained for the concentrated (black) and diluted (red) aqueous DA solutions. (**b**) The fluorescence emission (dashed lines) and excitation (continuous lines) spectra obtained for the aqueous DA solution with different DA concentrations.

**Figure 2 ijms-23-05483-f002:**
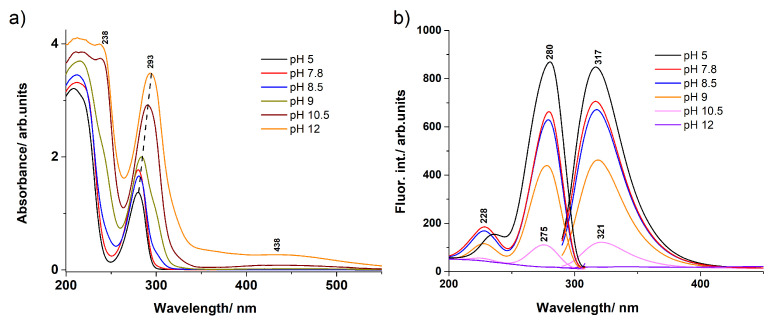
(**a**) The experimental UV absorption spectra obtained for aqueous DA solutions prepared having different pH values of the solution. (**b**) The fluorescence emission and excitation spectra obtained for aqueous DA solutions with different pH values.

**Figure 3 ijms-23-05483-f003:**
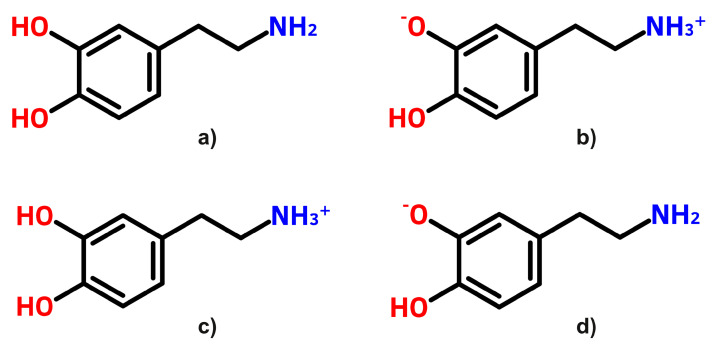
The chemical structure of the four dopamine conformations: (**a**) standard (or DA), (**b**) zwitterionic (or DA${}^{zw}$), (**c**) protonated (or DAH${}^{+}$), and (**d**) deprotonated (or DAH${}^{-}$), respectively.

**Figure 4 ijms-23-05483-f004:**
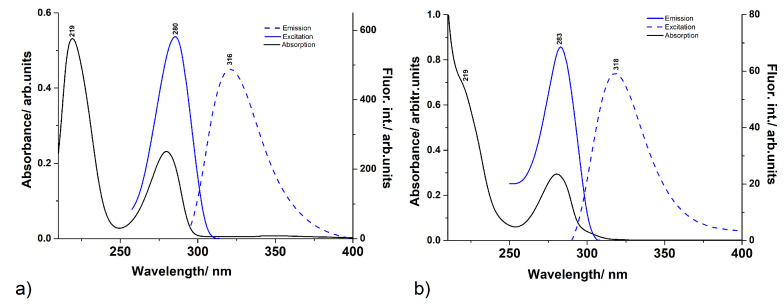
UV–Vis absorption spectra together with the fluorescence emission and excitation spectra recorded from dopamine solutions at pH 5.5 (**a**) and pH 8 (**b**), respectively.

**Figure 5 ijms-23-05483-f005:**
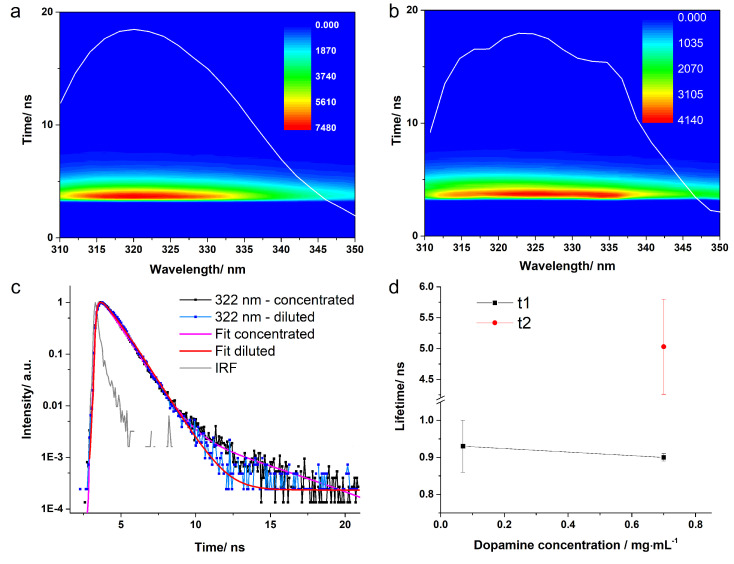
Time-correlated single-photon counting (TCSPC) 3D data carpet obtained from the concentrated (**a**), respectively diluted (**b**) DA solution at pH = 5.5 following excitation at 280 nm. The white line represents the integrated fluorescence spectrum. Normalized kinetic traces recorded at 322 nm with the fitted curves (**c**). The average lifetimes obtained for each solution (**d**).

**Figure 6 ijms-23-05483-f006:**
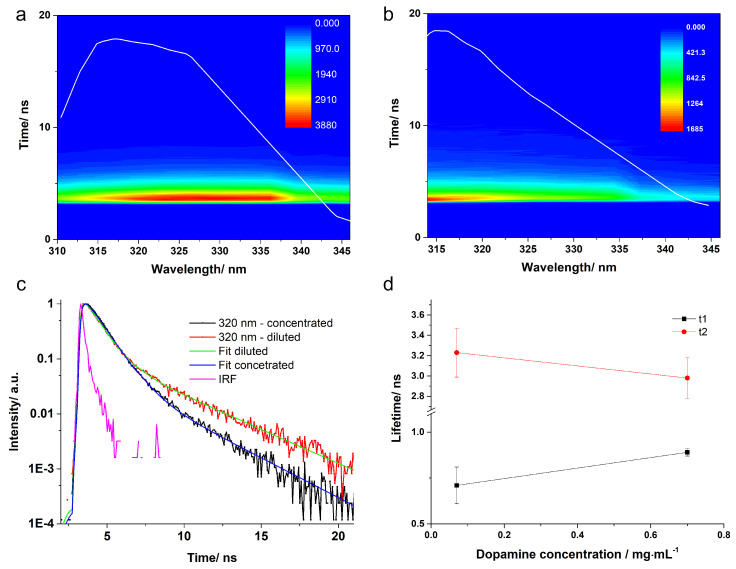
Time-correlated single-photon counting (TCSPC) 3D data carpet obtained from the concentrated (**a**), respectively diluted (**b**) DA solution at pH = 8 following excitation at 280 nm. The white line represents the integrated fluorescence spectrum. Normalized kinetic traces recorded at 320 nm with the fitted curves (**c**). The average lifetimes obtained for each solution (**d**).

**Table 1 ijms-23-05483-t001:** Vertical electronic transitions (in nm) and their oscillator strengths in the spectral range of 200–300 nm found for the DA, DA${}^{zw}$, DAH${}^{+}$, and DAH${}^{-}$ equilibrium geometries obtained at the DLPNO-STEOM-CCSD/ma-def2-TZVPP level of theory.

State	Geometries										
	**DA**			**DA** ${}^{\mathrm{zw}}$			**DAH** ${}^{+}$			**DAH** ${}^{-}$	
	ω	f		ω	f		ω	f		ω	f
S${}_{1}$	266	0.0275		292	0.0014		270	0.0379		302	0.0009
S${}_{2}$	226	0.0024		286	0.0803		222	0.0011		287	0.0858
S${}_{3}$	208	0.0637		249	0.0008		209	0.0378		255	0.0033
S${}_{4}$				242	0.0033					250	0.0164
S${}_{5}$				239	0.0088					230	0.0947
S${}_{6}$				224	0.0108					226	0.1094
S${}_{7}$				216	0.0119					219	0.0050
S${}_{8}$				214	0.0034					216	0.0018
S${}_{9}$				211	0.0045					216	0.0107
S${}_{10}$				209	0.0351					211	0.0002

**Table 2 ijms-23-05483-t002:** The parameter fits (fluorescence lifetime components, errors, relative amplitude, and amplitude average lifetime) calculated using the reconvolution method for the concentrated, respectively diluted, solutions at both pH values.

pH	Solution	Reconv. Method	$\tau_{1}$ (ns)	Error	Ampl. (%)	$\tau_{2}$ (ns)	Error	Ampl. (%)	$\tau_{\mathrm{avr}.}$ (ns)	Error
5.5	Concentrated	1-exp	0.89	0.01	100	-	-	-	-	-
		2-exp	0.91	0.02	99.58	4.86	0.38	0.42	0.99	0.03
	Diluted	1-exp	0.93	0.07	100	-	-	-	-	-
8.0	Concentrated	2-exp	0.89	0.01	95.83	2.98	0.20	4.17	0.97	0.05
	Diluted	2-exp	0.71	0.10	88.97	3.23	0.24	11.03	1.56	0.15
9.0	Concentrated	2-exp	0.69	0.02	92.61	2.41	0.20	7.40	0.82	0.01
	Diluted	2-exp	0.64	0.10	88.20	2.96	0.18	11.80	0.91	0.11

**Table 3 ijms-23-05483-t003:** Fluorescence lifetimes (in seconds) defined as the inverse of the radiative transition rate obtained for the four DA species at the ωB97X-D3/ma-def2-TZVPP level of theory. (The contribution of Franck–Condon and Herzberg–Teller approximations is given in parenthesis.)

	DA	DA${}^{\mathrm{zw}}$	DAH${}^{+}$	DAH${}^{-}$
FC ${}^{a}$	4.39 × 10${}^{-9}$	8.47 × 10${}^{-7}$	4.70 × 10${}^{-9}$	1.36 × 10${}^{-6}$
	(100%)	(100%)	(100%)	(100%)
FC+HT ${}^{b}$	3.79 × 10${}^{-9}$	2.75 × 10${}^{-9}$	4.51 × 10${}^{-9}$	1.14 × 10${}^{-8}$
	(86%, 14%)	(0%, 100%)	(96%, 4%)	(1%, 99%)
FC+(HT+D ${}^{c}$)	3.16 × 10${}^{-9}$	1.02 × 10${}^{-9}$	2.95 × 10${}^{-9}$	7.10 × 10${}^{-9}$
	(81%, 19%)	(0%, 100%)	(78%, 22%)	(0%, 100%)

${}^{a}$ Franck–Condon approximation. ${}^{b}$ Herzberg-Teller approximation. ${}^{c}$ Duschinsky rotation effects.

**Table 4 ijms-23-05483-t004:** The *QY* of dopamine at different pH values and concentrations calculated under 280 nm excitation.

pH	Solution	*QY* (%)
5.5	Concentrated	3.4
	Diluted	3.4
8.0	Concentrated	2.0
	Diluted	2.4

## Data Availability

Data available in a publicly accessible repository.

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
