# Peer review of "Dopamine Photochemical Behaviour under UV Irradiation"

_ijms, 2022, doi:10.3390/ijms23105483_

Round 1

Reviewer 1 Report

The work of Flamas et al explores the spectroscopic behaviour of dopamine as a function of concentration and pH. Both experimental and theoretical results are presented. Although the methods employed by the authors are valid, there are several inaccuracies in the interpretation of the results, which require further evaluation before publication.

  • Figure 1 presents the absorption, excitation and emission spectra of concentrated and diluted dopamine solution, but the pH is not specified. The authors interpret the excitation spectrum of the concentrated solution based on their calculations, suggesting that both the neutral and zwitterionic form are present in solution. Which is the prevailing species in the diluted solution according to the authors? Given the similarity of the emission spectra in both cases, do they assume that emission is mostly coming from the same species in the two cases? Which is the emissive species in the to cases?
  • The spectra reported in Figure 3 are not in agreement with the description reported in the text. The authors indeed state that a reduction of absorption is observed in case of pH=8, which is not noticed in the figure. Furthermore it is stated that fluorescence is red shifted at pH=8, with its maximum moving from 316 nm at pH=5 to 318 nm at pH=8, while in the figure the fluorescence maximum is marked as 321 nm at pH=5 and 317 nm at pH=8.
  • Which is the concentration of the solutions at pH=5 and pH=8 used to produce figure 3?
  • The authors also state that the dominant species at pH=5 is DAH+ while at pH=8 it is DAH-: on which basis they make this assumption? Did they determine the pKa of dopamine for its multiple possible deprotonation events?
  • The weight of the fluorescence lifetimes obtained by the fit should be reported in Table 2.
  • In order to obtain more solid evidences about the pH influence on the spectral properties of dopamine, measurements should be repeated at different pHs, spanning a more extended interval.
  • Caution should be taken when interpreting fluorescence spectra and corresponding fluorescence lifetimes measured at high concentration, since several artefact could invalidate the results (inner filter effect, aggregate formation etc)

Author Response

Please, see our responses in the attached PDF file 

Reviewer 2 Report

The manuscript investigates the origin of optical and photophysical behavior of dopamine from a molecular level perspective. The manuscript is well-written and claims are reasonably well-supported by the experimental and theoretical data. The manuscript, in its current form, however, does not provide much insights into the excited state photophysics of dopamine.

I have following comments on the manuscript,

  1. In concentrated solution, intermolecular H-bonds amongst different dopamine molecules will have a significant impact on the absorption and fluorescence property.  Do they lead to aggregate formation? How valid is, then, to rationalize the experimental data on the basis of different individual conformation of DA (protonated, unprotonated, zwitterionic etc)? Note that this situation is different in dilute solution, where H-bonding with the solvent will have a major role in the deactivation pathway of dopamine. Can authors comment on these issues?
  2. The experimentally measured absorption peak maxima for the protonated (DAH+) and deprotonated (DAH-) dopamine differ considerably from the theoretically calculated value (shown in Table 1). What is the origin of such discrepancy? Given such mismatch between the experimental and theoretical absorption peak maxima, how reliable are the inferences made from the calculated oscillator strength?
  3. The theoretically calculated fluorescence peak values differ considerably from the experimental values for DAH+ and DAH-. Can authors comment on the limitation of the theoretical models employed here?
  4. Does the theoretical calculations provide any information about the normal modes responsible for the excited state decay?

Author Response

Please, see our responses in the attached PDF file.

Round 2

Reviewer 1 Report

The authors addressed almost all the previous concerns. My only remark is that also in the revised version the absorption spectra reported in Figure 4 still do not reflect the  statement reported on page 7 for the part regarding the change in absorption intensity:

'Both absorption and excitation spectra showed a peak around 280 nm that was slightly shifted to higher  wavelengths and decreased in intensity about 4 times for the pH 8 solution compared to pH 5.5 solution.'

Author Response

Thank you very much for your. comments, the wrong wording has been corrected. See rows: 226-228.

Yours, Sincerely

Attila Bender

Reviewer 2 Report

I thank the authors for satisfactorily adressing all my concerns. I recommend the manuscript for publication.

Author Response

Thank you very much for your useful comments.

Yours, Sincerely

Attila Bende